# The Magnetosome Protein, Mms6 from *Magnetospirillum magneticum* Strain AMB-1, Is a Lipid-Activated Ferric Reductase

**DOI:** 10.3390/ijms231810305

**Published:** 2022-09-07

**Authors:** Dilini Singappuli-Arachchige, Shuren Feng, Lijun Wang, Pierre E. Palo, Samuel O. Shobade, Michelle Thomas, Marit Nilsen-Hamilton

**Affiliations:** 1Ames Laboratory, U.S. Department of Energy, Ames, IA 50011, USA; 2Aptalogic Inc., Ames, IA 50014, USA; 3Roy J. Carver Department of Biochemistry, Biophysics and Molecular Biology, Iowa State University, Ames, IA 50011, USA

**Keywords:** Mms6, ferric reductase, lipid, bicelles

## Abstract

Magnetosomes of magnetotactic bacteria consist of magnetic nanocrystals with defined morphologies enclosed in vesicles originated from cytoplasmic membrane invaginations. Although many proteins are involved in creating magnetosomes, a single magnetosome protein, Mms6 from *Magnetospirillum magneticum* strain AMB-1, can direct the crystallization of magnetite nanoparticles in vitro. The in vivo role of Mms6 in magnetosome formation is debated, and the observation that Mms6 binds Fe^3+^ more tightly than Fe^2+^ raises the question of how, in a magnetosome environment dominated by Fe^3+^, Mms6 promotes the crystallization of magnetite, which contains both Fe^3+^ and Fe^2+^. Here we show that Mms6 is a ferric reductase that reduces Fe^3+^ to Fe^2+^ using NADH and FAD as electron donor and cofactor, respectively. Reductase activity is elevated when Mms6 is integrated into either liposomes or bicelles. Analysis of Mms6 mutants suggests that the C-terminal domain binds iron and the N-terminal domain contains the catalytic site. Although Mms6 forms multimers that involve C-terminal and N-terminal domain interactions, a fusion protein with ubiquitin remains a monomer and displays reductase activity, which suggests that the catalytic site is fully in the monomer. However, the quaternary structure of Mms6 appears to alter the iron binding characteristics of the C-terminal domain. These results are consistent with a hypothesis that Mms6, a membrane protein, promotes the formation of magnetite in vivo by a mechanism that involves reducing iron.

## 1. Introduction

Since first reported in 1975 [1] magnetotactic bacteria (MTB) have attracted interest because of their abilities to synthesize magnetite crystals in specialized organelles called “magnetosomes” [2,3,4]. Gene regulation and genomic analysis related to magnetosome formation have been extensively studied [5,6,7]. Superparamagnetic magnetite crystals of similar size and shape to the bacterial magnetites are formed in vitro due to the presence of recombinant Mms6, a magnetosome-associated protein [8,9,10]. Although Mms6 alone is not responsible for the formation of magnetic nanoparticles in vivo [11], its in vitro activity provides us an opportunity to better understand the mechanism by which this biomineralization protein functions. Such knowledge helps us to understand how magnetotactic bacteria can synthesize the magnetic crystals in magnetosomes and informs the design of bio-inspired routes to synthesize iron oxides and other studies related to magnetic nanoparticles [12,13,14,15,16,17,18].

We have previously demonstrated that Mms6 forms a micellar quaternary structure in vitro that may provide a surface for magnetite nanoparticle formation [9]. Mms6 consists of two subdomains, with the N-terminal domain responsible for anchoring the C-terminal domain in the micelle from which the C-terminus binds iron and forms magnetic nanoparticles in vitro in the presence of high concentrations of Fe^2+^ and Fe^3+^. Analysis of Mms6 and its synthetic C-terminal domain by fluorescence and CD spectroscopy provided evidence that the protein undergoes a structural change upon binding iron and exhibits two modes of interaction with iron [9,19].

The magnetite crystal lattice contains Fe^2+^(Fe^3+^)_2_O_4_. However, Mms6 binds Fe^3+^ much more effectively than Fe^2+^ [20]. Curiously, the affinity of the isolated C-terminal for these two oxidation states of iron is the reverse of the holo-protein [21], which suggests that the N-terminal domain influences the structure and function of the C-terminal domain.

Although the lower affinity of Mms6 for Fe^2+^ can be circumvented in vitro by making a high concentration of Fe^2 +^available, the ratio of Fe^3+^/Fe^2+^ in vivo is unlikely to be 2:1 as provided in vitro. Rather, Fe^3+^ is proposed as the predominant form of iron in magnetosomes [22,23,24,25,26,27]. Thus, if Mms6 were to be involved in initiating or promoting the growth of magnetite crystals in vivo, it would need to cooperate with a protein that could reduce the available Fe^3+^ to provide sufficient Fe^2+^ for crystal growth or itself be a reductase.

Here we show that Mms6 is a ferric reductase capable of producing the Fe^2+^ required as a component of the magnetite crystal lattice. Mms6 shows structural homology with the ferric reductase superfamily, but it does not require the presence of a heme group to reduce iron. The results of structural homology and mutational analysis suggests that the reductase catalytic site is in the N-terminal domain. Consistent with its association with magnetosome membranes when isolated from cells and other evidence that it is membrane-localized in vivo [8,11,28], we show that the reductase activity of Mms6 is enhanced when the protein is integrated in a lipid bicelle membrane. Thus, we propose that the function of Mms6 in vivo contributes to both essential elements of magnetic crystal formation: the reduction of iron and the assembly of Fe^2+^ and Fe^3+^ into the crystal structure.

## 2. Results

### 2.1. Mms6 Is a Ferric Reductase

The results of our previous studies suggested that Mms6 binds Fe^3+^ cooperatively in groups of three [9] and binds Fe^2+^ poorly [20]. These binding characteristics are not compatible with an independent role of Mms6 in building the crystal lattice of magnetite, which contains Fe^3+^:Fe^2+^ at a ratio of 2:1. We reasoned that, if Mms6 has a direct role in building magnetite in vivo, it should be capable of reducing Fe^3+^ to create the Fe^2+^ necessary for building the crystal lattice. Reduction of Fe^3+^ to Fe^2+^ by Mms6 was monitored by the increase of A_562_ from the Fe^2+^-ferrozine complex. With this assay, we found that Mms6 can reduce Fe^3+^ to Fe^2+^ under aerobic (Figure 1A,B,D) and anaerobic (Figure 1B,C) conditions. To establish that the observed activity is likely due to catalysis by Mms6, we tested five sequence variants of the protein, which were m1Mms6 (a shuffle of the charged residues in the C-terminal 21 amino acid residues), m2Mms6 (a shuffle of the 9 -OH and -COOH containing amino acids in the C-terminal 21 residues), m3Mms6 (shuffle of the C-terminal 21 amino acid residues), Mms6 (W119A), and Mms6-GLtoGA (L108A, L110A, L112A, L114A, L116A), all of which demonstrated decreased reductase activity (Figure 1A,D). A key to the position numbers for Mms6 is found in Appendix A and the sequences of recombinant Mms6 and mutants are found in Appendix A. These mutations cover the length of the protein, which consists of two domains characterized by their hydrophobicity, with the N-terminal being hydrophobic and the C-terminal hydrophilic (Appendix A).

We have previously demonstrated that Mms6 forms micelles in vitro. A trivial reason that mutant and scrambled forms of Mms6 lack reductase activity would be their aggregation or otherwise structural impairment at the macromolecular level. To minimize possibilities for variations in the gross quaternary organization of Mms6, all mutations, and scrambled versions of Mms6 were chosen to maintain a similar hydropathy plot (Appendix A). This expectation was confirmed by previous analysis of these mutations that demonstrated their abilities to form micelles [19] and TEM images of Mms6 and its mutants, which show similar morphologies (Appendix A). Thus, it is unlikely that the mutant versions of Mms6 lack reductase activity due to reorganized quaternary structure. The lack of reductase activity with mutation of Mms6 supports the notion that the reductase activity displayed by Mms6 requires a defined catalytic site.

### 2.2. Electron Donor and Cofactor Requirement

To determine its requirements for electron donor and co-factor, Mms6 was tested for reductase activity in the presence of combinations of electron donors and cofactors. The results show that Mms6 prefers NADH and can also use NADPH as electron donor but exclusively uses FAD over FMN as cofactor (Figure 2A left). The synthetic C terminal domain, C21Mms6, was also tested under the same aerobic conditions as for Mms6, and no significant activity was observed (Figure 2A right). The K_M_ for FAD and NADH were determined as ~25 and ~15 µM by their abilities to stimulate reductase activity, and similar activity isotherms were obtained for FAD under aerobic and anaerobic conditions (Appendix A). The addition of heme had no effect on the reductase activity of Mms6 (Appendix A).

The K_M_ of Mms6 for Fe^3+^-citrate is 50 µM, whereas BSA present in the same assay gave no activity (Figure 2B). In the presence of saturating Fe^3+^ and 100 µM each of FAD and NADH, the specific activity was estimated as 0.24 ± 0.20 nmole/min/mg (N = 20) and the k_cat_ as 4.8 × 10^−5^ ± 5.2 × 10^−5^ s^−1^. The affinity for Fe^3+^ was investigated under aerobic and anaerobic conditions for FeCl_3_ and under aerobic conditions for ferric citrate. The K_d_ for FeCl_3_ under aerobic and anaerobic conditions were 74 and 76 nM, respectively, whereas the K_d_ for ferric citrate was significantly higher at 14 µM (Figure 2C). This difference is to be expected as citrate has a high affinity for Fe^3+^ and reduces the concentration of free Fe^3+^ available for Mms6 binding.

### 2.3. Iron Binding Residues in Mms6

For reductase activity, Mms6 must bind and reduce Fe^3+^. To identify potential iron-chelating residues, we made a series of alanine exchange mutants for each -OH or -COOH containing sidechain (S, D or E) in the C-terminal domain. We have previously demonstrated that the C-terminal mutant m2Mms6 (Appendix A) binds Fe^3+^ with a very low affinity that appears as nonspecific [9,19]. This and other Mms6 mutants were tested for iron binding and ferric reductase activities (Figure 3). For both assays, the activities of the mutants were normalized to that of the wild-type protein. The effects of mutations on reductase and iron binding activity were highly correlated with mutations at positions S138, S143, and S146 and their combinations showing large decreases in iron binding and reductase activities (Figure 3B and Appendix A). Although E148A had wild-type activity, the Mms6 (S146A, E148A) double mutant showed complete loss of both iron binding and reductase activities. The concordance in the results from these two assays strongly suggests that S138, S143, and S146 might be important for iron binding, which is required for reductase activity of Mms6. However, this observation does not rule out the possibility that one or more of these residues plays a structural role for one or both activities.

### 2.4. Mms6 Primary Sequence and Tertiary Structure Predictions

To better understand the relation between Mms6 and other prokaryotic proteins, the primary sequence of Mms6 from *Magnetospirillum magneticum*, AMB-1 used in this study was compared with sequences of Mms6 from related organisms and with other proteins (Mms7 and Mms13) that were also found tightly associated with magnetosomes [8]. The region of highest identity was the segment of GL repeats and W119 (Figure 4A).

We previously predicted a structure for Mms6 [19] using I-TASSER [29] and TM-align [30] to identify related proteins and protein families. This comparison identified homology with reductases (Appendix A). Here we developed a 3-dimensional model of Mms6 using SWISS-MODEL and AlphaFold2 [31,32,33], which identified the Photosystem II reaction center protein, W, as the most likely structural equivalent for Mms6. Based on the 3D model, the catalytically active sites in Mms6 were predicted using CASTp [34]. This analysis identified residues in the N terminal (W119 and the GL repeat) as the most likely sites to define the Mms6 catalytic domain (Figure 4B). These findings are consistent with our observations that mutations in these positions of the N-terminal domain result in loss of catalytic activity (Figure 1D).

Mms6 exists in vitro as micelles in a range of sizes; the smallest and most abundant size have an average diameter of 15 nm (Appendix A). These structures, which can be considered as being the minimal micellar size, aggregate in a linear fashion as seen in Appendix A. To further understand the quarternary structure of Mms6, we attempted to model it with Alphafold2 (Appendix A). The results show a growing structure that might align at an air–water interface as we observed on a Langmuir trough [20]. However, in aqueous solution, this structure must fold into itself to form a micelle with the hydrophobic N-terminal domain surrounded by the hydrophilic C-terminal domain. In previous studies, we predicted that the 15 nm diameter micelles observed in Appendix A might contain 20–40 Mms6 monomers based on the surface area occupied by each monomer as calculated from studies in a Langmuir trough [9]. If the micelles are considered to be solid spheres of Mms6, each is predicted to contain about 150 Mms6 molecules based on the reported density (~1.47 ng/cm^3^) of 10 KDa proteins [35]. The confidence level of quarternary structures predicted by Alphfold2 that contained more than two monomers was below the 50% cutoff for pLDDT (Appendix A), which suggests that the quaternary structure of the micelles is disordered.

### 2.5. Mms6 Integrates into Lipid Membranes

The N-terminal domain of mature Mms6, an amphipathic protein, is largely hydrophobic and might integrate into a membrane. Numerous observations also suggest that Mms6 is a membrane protein [8,9,20,28]. Membrane integration was tested using DLS and intrinsic fluorescence. Mms6 exists in solution as micelles with a hydrodynamic diameter of ~13 nm (Figure 5A, green). This result matches well with the average 15 nm diameter determined from negatively stained samples by TEM (Appendix A). Incubation with 0.5% of Triton-X100 at 24 °C followed by detergent removal with hydrophobic beads increased the Mms6 micellar hydrodynamic diameter to an average of ~45 nm, which we suspect is due to fusion of micelles induced by the detergent (Figure 5A, dashed cyan). DMPC/DHPC bicelles (25 mM) had hydrodynamic diameters of ~10 nm (Figure 5A, red), which is consistent with their expected size [36]. The incorporation of Mms6 into bicelles was not evidenced by a change in size of the bicelles as they had a hydrodynamic diameter of ~10 nm in the presence or absence of Mms6. However, incorporation of Mms6 into the bicelles is supported by the lack of particles the size of the Mms6 micelles (d = 12–15 nm) in the Mms6–bicelle sample. This lack of apparent size difference between the bicelles alone or with Mms6 could occur if incorporation of Mms6 into the bicelles resulted in a change in shape from discoidal to spherical, which might not be observed as a change in hydrodynamic diameter as the analysis of DLS data assumes a spherical shape for all particles. A small change in hydrodynamic diameter was observed for liposomes with the loss of the Mms6-Triton X-100 peak, again suggesting integration of Mms6 into these membranes (Figure 5A, blue).

To further evaluate the incorporation of Mms6 into the bicelles, we measured intrinsic fluorescence. When excited at 290 nm, Mms6 shows a fluorescence spectrum with a λ_max_ at 355 nm, which shifts to 346 nm when the protein is integrated into bicelles. This blue shift is consistent with the interpretation that tryptophan is experiencing a more hydrophobic environment, which is expected after its integration into lipid membranes. To identify the Trp residue responsible for the blue fluorescence shift, we tested two Mms6 mutants (W103F and W119F). The blue shift was observed with the W103F mutant but not with the W119F mutant (Figure 5B and Appendix A). These results suggest that W119, but not W103, adopts a new, more hydrophobic environment when Mms6 is integrated with bicelles.

### 2.6. Lipids Promote Higher Ferric Reductase Activity of Mms6

When incorporated into bicelles or liposomes, the ferric reductase activity of Mms6 was higher by an average of two-fold in liposomes (N = 2) and five-fold in bicelles (N = 2) (Figure 6A,B). The C and N terminal mutants of Mms6, which were inactive in the absence of bicelles (Figure 1), were also inactive when incorporated in bicelles (Figure 6C). Thus, these mutations identify critical amino acid residues in Mms6 that are required for reductase catalysis regardless of its environment.

### 2.7. Reductase Activity of the Mms6 Ubiquitin Monomer

To evaluate the role of its multimeric state on Mms6 activity, we reasoned that the protein could be retained as a monomer if fused at the N-terminus to a hydrophilic monomeric protein that does not form multimers. Consequently, we created fusion proteins of Mms6 and Mms6 (S146A, E148A) with ubiquitin. The monomeric structures of these fusion proteins were validated by size exclusion chromatography (Figure 7A). Whereas the reductase activity was the same in the monomeric as the multimeric state for the wild-type and mutant forms, the iron binding activity of Mms6 (S146A, E148A) was significantly lower than for Mms6, but the binding capacity for Ubi-Mms6 (S146A, E148A) was similar to that of Ubi-Mms6 and Mms6 (Figure 7B,C).

## 3. Discussion

**Mms6 in vivo function:** Identified from the isolated magnetosome membrane of AMB-1 as a magnetite-associated protein [8], Mms6 promotes the formation of magnetic nanoparticles in vitro when included in co-precipitation synthesis reactions [8,37]. Genetic evidence suggests that Mms6 regulates the morphology of magnetites in the later stage of crystallization in vivo and alternatively that it is an accessory protein, unnecessary for magnetite formation [38]. It has been proposed to alternatively hold certain proteins such as Mms5, Mms7, and Mms13 to the magnetite and function with them to impart the cubo-octahedral shape of magnetite crystals in AMB-1 [11], and it has been specifically assigned the role of promoting crystal growth on the 110 face of magnetite [28].

**Mms6 in vitro formation of magnetite:** Recombinant Mms6 binds iron with high affinity and high capacity and self-assembles into multimeric micelles that appear to be important for its in vitro function of promoting magnetite formation [9,19]. Mms6 binds Fe^3+^ much more effectively than it binds Fe^2+^ [20], which brings up the question of how Mms6 might interact with both Fe^2+^ and Fe^3+^ in the magnetosome to accomplish its assigned role of controlling growth at the 110 crystal surface.

**Mms6 reductase activity:** Here we have for the first time demonstrated that Mms6 is an iron reductase that uses FAD and NADPH as cofactors. Several protein structural prediction algorithms identified reductases by homology modeling and predicted active site residues in Mms6 (W119 or the GL string of residues) for which alanine substitution Mms6 mutants had minimal to no reductase activity. Thus, the combination of modeling and experimental analysis identified the N-terminal half of the protein as the likely reductase catalytic site. The GLGL segment and Trp 119 are conserved in a group of proteins related to Mms6 that are found attached to magnetosomes [8] and also links Mms6 to the ferric reductase superfamily (FRD, [39]). The presence of these features in other related proteins suggests that other members of the magnetosome protein family may also possess iron reductase activity.

**C-terminal Mms6 amino acids for iron binding:** We have previously demonstrated that the C-terminal domain of Mms6 binds iron and, when provided with Fe^2+^ and Fe^3+^, it can direct the crystallization of magnetite as can the full-length protein [9]. Consistent with our findings here that alanine substitutions of several C-terminal domain residues result in lower Fe^3+^ binding capability of Mms6, Rawlings and coworkers reported E142, E148 and R153 are key C terminal residues involved in Mms6 ferric binding. In addition, their simulation studies of a model DEEV peptide identified E148 and E149 as specifically binding to Fe^2+^ [40]. However, we show here that mutation of either amino acid residue to alanine did not alter reductase or iron binding activity of Mms6.

**Iron binding correlates with reductase activity for Mms6 C-terminal mutants, but not N-terminal mutants:** We found a correlated drop in reductase activity and iron binding activity with alanine substitutions at positions 138, 143, 146, and 148, all in the C-terminal domain. As the rate of catalysis depends on its ability to obtain Fe^3+^, which is mediated by the C-terminal domain, it is reasonable to consider that a decrease in reductase activity might parallel the loss of iron binding capability. However, the observed loss of reductase activity with the GL mutant, which retained iron-binding capability, identified the catalytic site as likely to be in the N-terminal domain.

**The C and N-terminal domains interact structurally and functionally:** We have previously shown that the binding of iron by the C-terminal domain in the multimer results in a change in the CD spectrum, intrinsic fluorescence, and SANS intensity profile, which suggests that a structural change in the C-terminal domain upon binding iron is transmitted to the N-terminal domain [19]. As an isolated peptide, the C-terminal domain showed no predominant structure by NMR analysis [21]. Yet a helical structure is predicted for the Mms6 C-terminal domain by Alphafold2 with less confidence in the C-terminal half of the domain, which is reflected in the fact that SWISS-MODEL did not propose a structure for this segment of the protein (Appendix A).

**The kinetic parameters of Mms6 are consistent with related enzymes:** At 6191 Daltons, Mms6 is the smallest of reported ferric reductases. However, if it exists in vivo as a complex with the remainder of its precursor protein, which is between 12,531 and 14,691 Daltons, depending on the translational start site, then Mms6 would reach the lower end of the molecular weight range of 13,000 Daltons reported for the *B. subtilis* ferric reductase ([41], Appendix A). A review of kinetic data for 18 reports of ferric reductases from various prokaryotes identified a range of specific activity values for the purified proteins (ferric citrate or ferric EDTA as acceptors) from 5 × 10^−3^ to 13,100 with a median value of 14 nmol/min/mg (Appendix A).

Our best estimates for the specific activity (0.24 nmole/min/mg) and k_cat_ (4.8 × 10^−5^ s^−1^, Appendix A) of Mms6 are lower than reported for most reductases. We suspect that this activity is lower than what would be observed in vivo, in part because Mms6 is refolded from an inclusion body during its preparation. Indeed, in two separate comparisons of the refolded protein with the ubiquitin fusion protein, which is isolated as a soluble monomer, the k_cat_ of the latter was 5 ± 0.2 times that of the refolded protein when assayed together on the same plate under the same assay conditions. Another likely reason for the low activity of Mms6 in vitro is that the protein is out of its natural lipid environment (discussed next), and it may be only part of a multisubunit holoenzyme that functions in vivo [42,43,44].

Except for its low activity, the functional characteristics of Mms6 are otherwise consistent with those of other reported reductases [45]. The pH optimum of 7 for the ferric reductase activity of Mms6 is the same as for most other reported ferric reductases (Appendix A). Mms6 prefers NADH over NADPH as electron donor and is specific for FAD as co-factor. Our estimated K_M_ for Mms6 of 50 μM for the iron-citrate is within the range of 6 to 213 μM reported for other reductases for which the average was 34 ± 64 μM (N = 10), median = 12 μM. However, our determined K_d_ for binding ferric citrate of 14 µM suggests that the K_M_ for enzyme activity is a more complex parameter than just depending on the binding of Mms6 with Fe^3+^-citrate and might be related to the affinities of FAD and NADH, which we estimated as ~25 and ~15 µM respectively.

**Integration into membranes increases Mms6 reductase activity:** Mms6 spontaneously integrates into liposomes [9] and orients at the surface of a Langmuir trough [20]. These data and those of others [8,9,20,28,46] supported the expectation that Mms6 would have an affinity for membranes. Mms6, which exists as micelles in the absence of lipid, was incorporated into bicelles, and the hydrodynamic diameters of the resulting particles were determined by dynamic light scattering. At concentrations as low as 0.2 mg/mL, Mms6 exists as micelles in aqueous solution at pH 3 or pH 7.5. At the latter pH, these micelles are characterized with diameters of ~12–15 nm [19]. We observed a specific change in the intrinsic fluorescence of W119 with no change in the fluorescence of W103 when Mms6 is integrated into bicelles (Appendix A). The shift is consistent with W119 becoming buried in a more hydrophobic location. This shift accompanies an increase in the reductase activity of Mms6 and suggests that the preferred environment of the reductase catalytic site is hydrophobic. However, with their simple and synthetic lipid composition, the bicelle environment is still far from the lipid composition of magnetosomes, and we anticipate higher activity of this enzyme in its native environment.

**The N-terminal reductase functions as a monomer, while the C-terminal iron binding characteristics vary with structural context:** While in the form of a multimer, the roles of individual Mms6 monomers cannot be distinguished, and it cannot be known if the Mms6 iron-binding and reductase activities depend on a multimeric structure or can be performed by a monomer. The ability to multimerize also adds complication to the interpretation of data for the effects of some of the mutations on reductase activity. For example, the GL to GA mutation and the W119 mutation destabilize the micelles resulting in a mixture of heteromeric and monomeric Mms6 [19]. To address the question of whether the reductase requires a multimeric structure, we created monomeric Mms6 by fusion to ubiquitin. This monomeric fusion protein and the S146, E148 mutant possessed the same relative reductase activity as the multimer, but mutations in the C-terminal domain that drastically decreased Fe^3+^ binding in multimeric Mms6 did not influence iron binding by the monomer. This data suggests that (1) the C-terminal domain but not the N-terminal domain structure and function is affected by the multimeric state of Mms6 and (2) the iron binding capability of the Ubi-Mms6 (S146, E148) monomer does not support catalysis.

The C-terminal domains interact with each other while associated in the micelle [9]. The results shown here support the hypothesis that the C-terminal domain adopts a “multimeric” structure in the micellar configuration that is different from its structure in the monomer. These alternative structures have similar capabilities to bind iron but may use different residues for capturing iron. This observation helps to explain the apparent contradiction between the higher affinity of Mms6 for Fe^3+^ [20] and the isolated C-terminal domain for Fe^2+^ [21] and our evidence that E148 and E149 do not contribute to Fe^3+^ binding by Mms6, yet these amino acid residues in the DEEV cluster are proposed to be central to binding Fe^2+^ by the isolated C-terminal 30 mer [21]. It must also be considered that S146 and E148 might not directly chelate iron only in the multimer but may be responsible for maintaining the C-terminal domain structure required for high affinity iron binding and the integration of Fe^3+^ capture with catalysis in the multimer.

**Summary:** We have demonstrated that Mms6 is an iron reductase with kinetic properties like other prokaryotic iron reductases and with a requirement for FAD and preference for NADH as cofactor and electron donor, respectively. The N-terminal domain of Mms6 contains the catalytic site that acts as a monomer, and the C-terminal domain binds iron. The structure of the C-terminal domain and its iron binding characteristic are influenced by its association with the Mms6 multimer. The reductase activity is enhanced by the incorporation of Mms6 into membranes. Based on these findings, we propose that the mechanism by which Mms6 contributes to magnetosome formation in vivo involves binding of Fe^3+^ by the C-terminal domain combined with reduction to Fe^2+^ by the N-terminal domain, which brings both forms of iron to the crystal surface that are necessary for growing the magnetite crystal lattice.

## 4. Materials and Methods

### 4.1. Reagents, Proteins, and Preparation of Mutants and Sequence Variants

Phospholipids used to make bicelles and liposomes were purchased as stocks dissolved in 100% chloroform from Avanti Polar Lipids (Alabaster, AL, USA). Bio-Beads™ SM-2 Resin used for removing detergents was purchased from Bio-Rad (Hercules, CA, USA). Other chemical reagents were of analytical grade or higher purity and were obtained from Sigma-Aldrich (St. Louis, MO, USA). Single site-directed mutagenesis of Mms6 was conducted using the QuikChange II mutagenesis kit from Agilent Technologies (Santa Clara, CA, USA) following the manufacturer’s instructions. C-terminal domain variants containing shuffled residues were created by replacing the appropriate segment of the cDNA with oligonucleotides encoding the desired amino acid sequences. All protein sequences used in this study can be found in Appendix A. Mms6 is a cleavage product of a longer primary sequence. The residues cited in this work are numbered relative to the proposed start site of the original translated protein based on the gene sequence (Appendix A). Position 1 in Mms6 is position 98 in the proposed primary sequence. The mature forms of Mms6 and its mutants were expressed and purified as described previously [9,10,19]. The C-terminal domain of Mms6 (C21Mms6: KSRDIESAQSDEEVELRDALA) was chemically synthesized by Genscript (Genscript Corp. (Piscataway, NJ, USA)).

### 4.2. Bicelle Preparation and Integration of Mms6 into Bicelles

Bicelle stocks consisting of 1,2-dimyristoyl-sn-glycerol-3-phosophocholine (DMPC) and 1,2-dihexyl-sn-glycero-3-phosphocholine (DHPC) (M_DMPC_:M_DHPC_ = 1:1, *q* = 1) with a total lipid concentration of 250 mM in buffer A (20 mM Tris, 100 mM KCl, pH 7.5 at 24 °C) were prepared as described in [36] with minor modifications. Chloroform was removed from an equimolar mixture of DMPC and DHPC in 100% chloroform in a glass vial on ice under a mild stream of argon in a ventilated hood. This lipid mixture was desiccated overnight under a constant vacuum at 4 °C. The desiccated lipids were resuspended in the appropriate volume of buffer A to achieve a total lipid concentration of 250 mM. The resuspended bicelle lipids were subjected to repeated warm (45 °C)/cool (ice) cycles until the solution became non-viscous and transparent. The bicelle (DMPC/DHPC) stocks were used immediately or aliquoted and stored at −20 °C until used.

The Mms6–bicelle complex was prepared as previously described [47]. A total of 80 micromolar Mms6 was mixed with 100 mM or 50 mM bicelle (*q* = 1) in buffer A. The test tubes were sealed with screw caps, and the protein–micelle mixtures were treated with four cycles of freeze (liquid nitrogen) and thaw (24 °C). The protein–bicelle mixtures were stored at 4 °C for up to three weeks or maintained at −20 °C before use. The Mms6–bicelles were incubated at room temperature for one hour before use for experiments.

### 4.3. Liposome Preparation and Integration of Mms6 into Liposomes

Five individual liposome stocks of 100 mM DMPC, 1-palmitoyl-2-oleoyl-*sn*-glycero-3-phosphocholine (POPC), 1,2-dioleoyl-sn-glycero-3-phosphocholine (DOPC), 1,2-dioleoyl-sn-glycero-3-phospho-(1′-rac-glycerol) (DOPG), or 1,2-dioleoyl-*sn*-glycero-3-phospho-L-serine (DOPS) in buffer A were prepared by extrusion through polycarbonate filters [48]. The Mms6-liposome stocks were prepared by mixing Mms6 and liposomes at final concentrations of 80 µM Mms6 and 8 mM liposome in buffer A with 0.5% Triton X-100 and incubating at 24 °C for 2 h with constant inversion. The Triton X-100 was removed by incubating with Bio-Beads™ SM-2 Resin pre-hydrated in buffer A at 24 °C with constant inversion for 3 h using a ratio of 35 µg Triton X-100 per mg of resin. The Mms6-liposomes were harvested by removing the supernatant after the beads were allowed to settle by gravity.

### 4.4. Ferric Reductase Activity

Ferric reductase activity was monitored by the spectral change in ferrozine (3-(2-Pyridyl)-5,6-diphenyl-1,2,4-triazine-p,p-disulfonic acid), which binds Fe^2+^ to form a complex with maximum absorbance at 562 nm and molar extinction coefficient of 27,900 M^−1^ cm^−1^ [49,50]. Unless otherwise stated, the assay was performed in a total volume of 100 or 200 μL with 20–40 μM Mms6 or Mms6 mutants in Buffer A (20 mM Tris-HCl, 100 mM KCl, pH 7.5), 0.1–0.2 mM NADH, 0.1–0.2 mM flavin adenine dinucleotide (FAD), and 0.8–1 mM ferrozine. Addition of ferric citrate initiated the reaction The same conditions were used to assay Mms6 integrated in bicelles (12.5 or 25 mM lipid) or liposomes (2 mM lipid). The reaction was started by the addition of ferric citrate and monitored by reading A_562_ each 1 or 2 min over 6 h. The background absorptions from samples lacking Mms6 read over the same time periods were subtracted from samples with Mms6 before calculating the rate of reductase activity. The V_max_, K_M_, and k_cat_ of Mms6 as a ferric reductase were fitted for K_M_ and V_max_ (A_max_) using the formula A = A_min_ + (A_max X_ S)/(S + K_M_) with Sigmaplot, where A = the change in absorption at 562 nm per min with A_min_ and A_max_ being the minimum and maximum values, respectively, and S = substrate concentration. All spectrophotometric measurements were performed at room temperature in 96-well plates (Falcon, Catalog# 351172 or 353948) and read with a Synergy II plate reader (Mms6 and mutants) or a Biotek, Model: Ceres 900 plate reader (C21Mms6). All determinations were performed independently at least twice in duplicate.

For each incubation time, mixtures containing the same components (excluding peptide or protein) as the assay mixture provided the blank values (averages of duplicates) that were subtracted from the average value obtained in the presence of protein or peptide. The concentration of Fe^3+^ was determined by A_562_ using an extinction coefficient of 27,900 M^−1^ cm^−1^. The reductase activity is expressed as nmol Fe^2+^/min/mmol protein. In some experiments, particularly those involving Mms6 integrated into bicelles, a significant lag was observed before enzymatic conversion was initiated. Biphasic kinetics has been observed for iron reductases by others [41,51,52,53] and variably attributed to the initial use of oxygen as the electron acceptor [52] or lower protein concentrations [53] or functioning in the absence of the larger protein complex with which it associates [41]. Consequently, for these occurrences, the initial velocities for Mms6 activity were taken after the initial lag period.

### 4.5. Iron Binding by Filter Capture

The binding isotherms of Mms6 for iron citrate and iron chloride binding were determined using ^55^Fe with either sodium citrate or sodium chloride to provide the appropriate anion. Binding was performed in 20 mM Tris-HCl, 145 mM NaCl, 0.01% Ac-BSA, 10 mM KCl, pH 7.0 and incubated for 1 h at 24 °C before filter capture (Figure 2) or in 20 mM Tris, 100 mM KCl, pH 7.5 and the reaction mixture incubated at 24 °C for 2 h before filter capture (Figure 3 and Figure 7). The reaction mixture was 100 µL (Figure 2) or 20 µL (Figure 3 and Figure 7), and the protein-^55^Fe complex was captured by filtration through a 0.45 µm (Millipore, Burligton, MA, USA, Cat#: HAWP 02500) followed by two 5 mL washes with the incubation buffer at 25 °C. The captured protein–iron complex was evaluated for ^55^Fe by scintillation spectroscopy in the range 0.0–18.6 meV.

### 4.6. Intrinsic Fluorescence Spectroscopy

A total of 20 micromolar Mms6 or a mutant Mms6 with or without 25 mM bicelles or 2 mM liposomes in buffer A were incubated at 24 °C for 2 h before collecting fluorescence spectra. Spectral analyses of the Trp-containing Mms6 and mutants were performed after subtracting the fluorescence readings of equimolar samples of the Trp-less Mms6 (W103A, W119A). The fluorescence spectrum of 40 µM tryptophan was obtained with and without bicelles in buffer A for correction of Mms6 spectra during decomposition of Mms6 fluorescence spectrum by the Protein Fluorescence and the Structure Toolkit [54].

### 4.7. Dynamic Light Scattering (DLS)

A total of 20 micromolar Mms6 and its mutants with or without bicelles or liposomes in buffer A were analyzed in Buffer A at 24 °C with a Zetasizer Nanoparticle analyzer (Model: ZEN3690, Malvern Instrument Ltd., Southborough, MA, USA). All samples were centrifuged at 14,000 × *g* at 24 °C for one hour to remove particulates prior to taking DLS measurements.

### 4.8. Size Exclusion Chromatography

Size exclusion chromatography was performed in an AKTA FPLC system (GE Healthcare, Chicago, IL, USA) with a prepacked Superose 12 10/300GL (separation range: 1 kDa to 300 kDa; GE Healthcare, Cat#17517301) at °C and a flow rate of 0.2 mL/min. The inner dimensions of the column were 10 × 300–10 × 310 mm with a bed volume of 24 mL. Prior to being loaded on the column, samples were dialyzed against the column buffer (20 mM Tris, 100 mM KCl, pH 7.9) then centrifuged at 15,900 RCF at 4 °C for 30 min. Blue dextran (300 kDa) was used to determine the void column volume (Vo), and the elution volumes (Ve) of ovalbumin (44.3 kDa), lysozyme (14.3 kDa), and aprotinin (6.5 kDa) were used to create a standard curve by which the apparent molecular weights of Mms6 and the ubiquitin-Mms6 fusion protein (Ubi-Mms6) were determined.

### 4.9. Analysis of Binding Isotherms and Statistical Evaluations

Where appropriate the Student’s T-test was used for comparison of data sets to determine statistical significance. The comparisons are noted in specific figures with the relevant *p* values identified by asterisks. The binding isotherms were analyzed using Sigmaplot with the formula B = B_min_ + (B_max_ x L)/(L + K_d_) [55]. All reported values for K_M_ and K_d_ passed the Normality (Shapiro–Wilk) and the Constant Variance (Spearman Rank Correlation) tests.

## Figures and Tables

**Figure 1 ijms-23-10305-f001:**
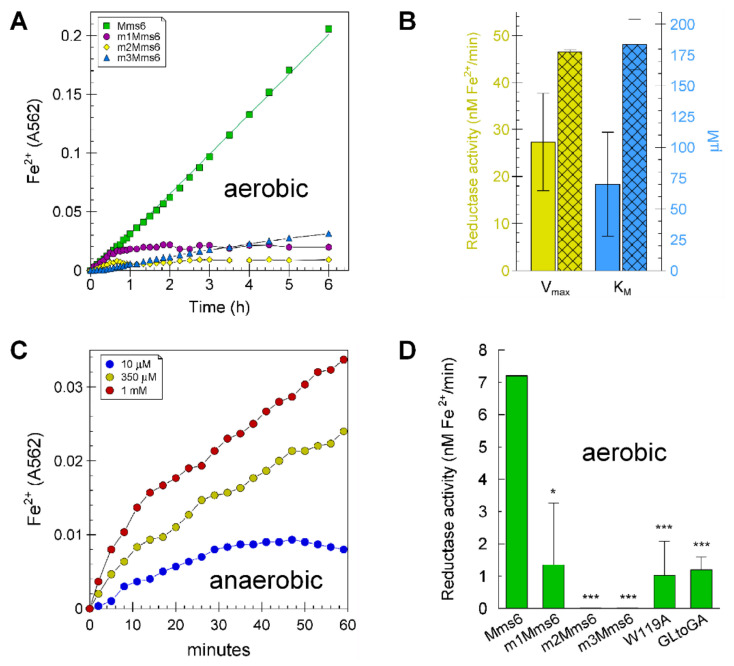
Reductase activity of Mms6. Ferric reductase activity was measured as described in Materials and Methods: (**A**) the reduction of 70 µM Fe^3+^-citrate by 20 µM Mms6, m1Mms6, m2Mms6, and m3Mms6 under aerobic (atmospheric) conditions in the presence of 100 µM each of NADH and FAD in Buffer A; (**B**) comparison of the Vmax (gold bars) and K_M_ (blue bars) for Mms6 under aerobic (empty bars) and anaerobic (cross-hatched bars) conditions; (**C**) ferric reductase activity of Mms6 (20 µM) at three Fe^3+^-citrate concentrations under anaerobic (argon) conditions in 100 µM NADH and 40 µM FAD in 70 mM Tris, 100 mM KCl, pH 7.5 and 1 mM ferrozine; (**D**) the reduction of 70 µM Fe^3+^-citrate by 20 µM Mms6, and various mutants under aerobic conditions. The asterisks denote *p* values of <0.05 (*) and <0.001 (***) in paired comparisons with Mms6.

**Figure 2 ijms-23-10305-f002:**
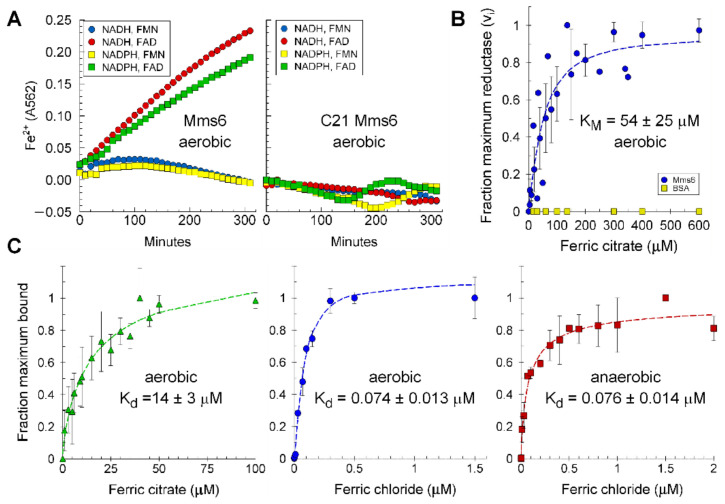
Mms6 ferric reductase specificity for cofactor and electron donor: (**A**) Mms6 (left) and C21Mms6 (right) were tested with 100 µM NADH or NADPH as electron donors combined with either 100 µM FAD or FMN as cofactors in 20 mM Tris, 100 mM KCl, 1 mM ferrozine, and 100 mM ferric citrate, pH 7.5. Reaction mixes run in parallel without protein were used as the background to subtract from the results of incubation with 20 µM Mms6 with the combinations of cofactor and electron donor shown; (**B**) reductase activity (initial velocity) was determined at a range of Fe^3+^-citrate concentrations to determine the K_M_ ± standard error for the reductase. Shown is the compiled data from 9 independent experiments involving 6 different protein preparations (r^2^ = 0.80, error of the estimate = 0.16); (**C**) the K_d_ of Mms6 was determined under aerobic conditions at pH 7.5 and pH 7 for ferric citrate and ferric chloride, respectively, and anaerobic conditions at pH 7 for ferric chloride. The ferric citrate binding isotherm was created from the averaged results of 11 independent experiments (duplicates per experiment) with 2 assessed by intrinsic fluorescence and the remainder by ^55^Fe binding with filter capture. In total, 5 different protein preparations were tested. The aerobic and anaerobic ferric chloride binding isotherms are each the average of two independent experiments with one (anaerobic) or two (aerobic) protein preparations and assessed by ^55^Fe binding with filter capture. On each plot, the estimated K_d_ ± standard error is shown. Goodness of fit: Fe-citrate, aerobic (r^2^ = 0.97, error of the estimate = 0.061), FeCl_3_, aerobic (r^2^ = 0.99, error of the estimate = 0.057), and FeCl_3_, anaerobic (r^2^ = 0.98, error of the estimate = 0.056).

**Figure 3 ijms-23-10305-f003:**
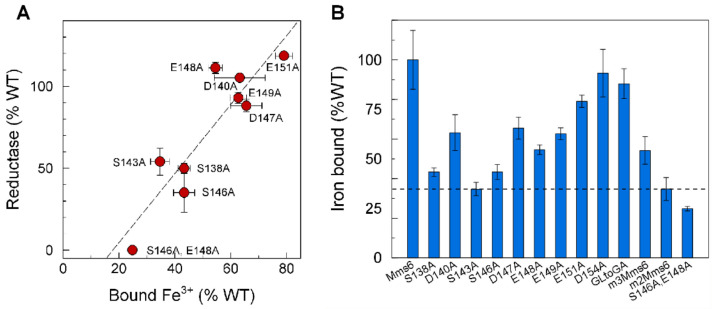
Iron binding and reductase activities are correlated: (**A**) the ferric reductase activity of each recombinant protein was determined as described in Materials and Methods with 20 µM protein and 100 µM ferric citrate; (**B**) the iron binding activity of each recombinant protein was determined as described in Materials and Methods with 1 µM protein and 20 µM ferric citrate. Reductase and iron binding activities were normalized to that of the wild-type protein. The dashed line identifies the binding activity of m3Mms6, which has been previously shown to bind Fe^3+^ nonspecifically [9].

**Figure 4 ijms-23-10305-f004:**
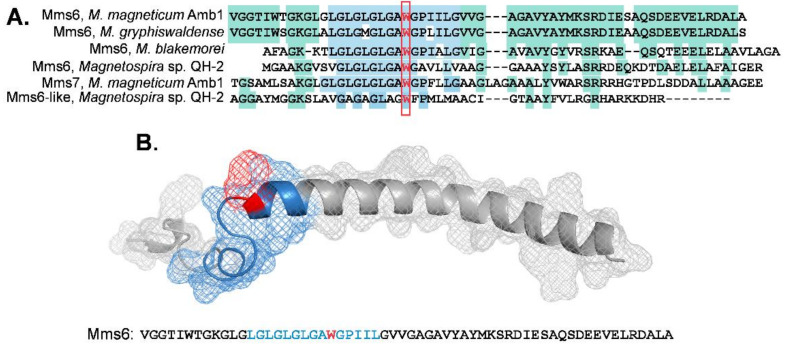
Primary and tertiary structural analysis of Mms6: (**A**) alignment of Mms6 and proteins with related sequence. Primary sequences were obtained from the NIH protein database (http://www.ncbi.nlm.nih.gov/protein, accessed on 12 June 2021) and aligned using the clustalw2 online server. The colored regions indicate identity of sequence with the blue region identifying the GL domain and the red rectangle the conserved tryptophan; (**B**) the AlphaFold2 predicted structure for Mms6 with the conserved Trp and GL domain identified and the primary sequence below. See Appendix A for 3D models of Mms6 predicted by other algorithms.

**Figure 5 ijms-23-10305-f005:**
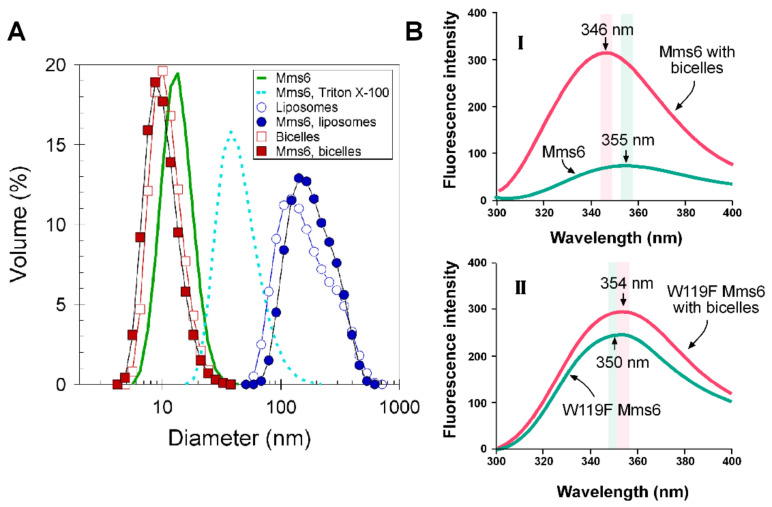
Mms6 interaction with lipids: (**A**) dynamic light scattering analysis of samples (500 µL in 20 mM Tris, 100 mM KCl, pH 7.5) of 25 mM bicelles, 2 mM liposomes or 0.5% Triton X100 with or without 20 µM Mms6 as stated in the legend on the figure. Size distributions for each sample are shown as a volume percentage; (**B**) fluorescence spectra were collected (λ_ex_ = 290 nm) and analyzed as described in Materials and Methods. Smoothed curves are shown for the fluorescence intensity of Mms6 (**I**) or W119F Mms6 (**II**). The vertical pink and green bars identify the positions of the maximum fluorescence intensity (±2.5 nm, based on a slit width of 5 nm) for each peak colored in red and green, respectively.

**Figure 6 ijms-23-10305-f006:**
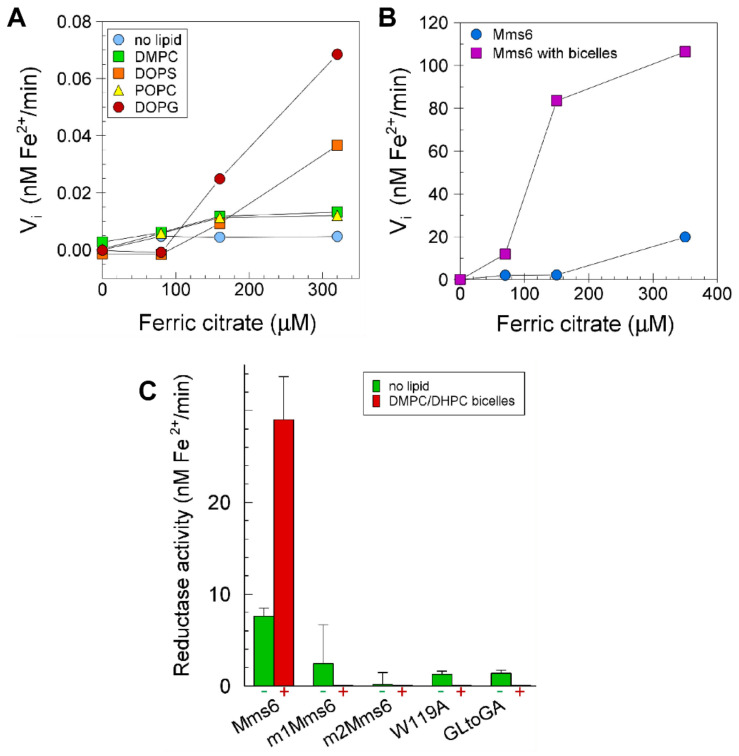
Mms6 has higher ferric reductase activity in lipid environments than in the absence of lipid: (**A**) Mms6, in liposomes of various lipid compositions as shown in the legend in the figure, was assayed in the presence of 100 µM each of NADH and FAD for ferric reductase activity with 0.8 mM ferrozine. The molar ratio of lipid:Mms6 was 100:1. The full chemical names of these lipids are defined in Materials and Methods; (**B**) Mms6 alone or incorporated into 25 mM q = 1 DMPC/DHPC bicelles was assayed in the presence of 200 µM each of NADH and FAD for reductase activity with 1.6 mM ferrozine; (**C**) the Vmax of Mms6 and identified mutants while free in solution (green bars) or associated with 12.5 mM bicelles (red bars) were determined in the presence of 100 µM each of NADH and FAD with 0.8 mM ferrozine. In all assays, Mms6 was 20 µM and the buffer was 20 mM Tris, 100 mM KCl, pH 7.5.

**Figure 7 ijms-23-10305-f007:**
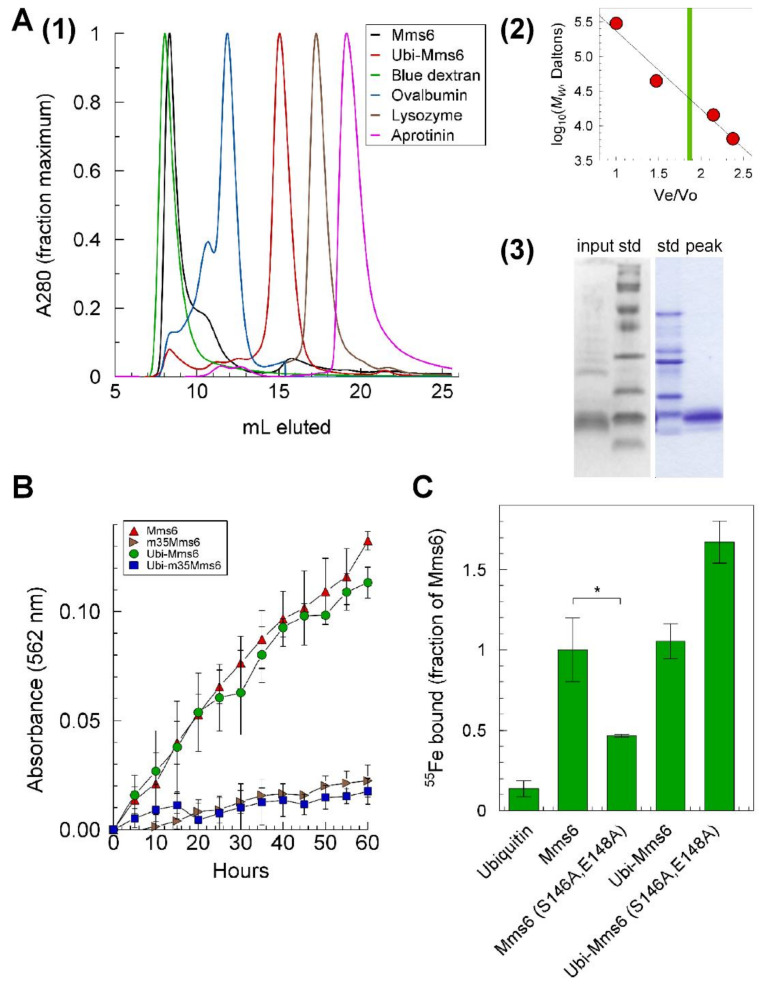
Monomeric Mms6 is a functional reductase but with altered C-terminal domain specificity for Fe^3+^: (**A**) (**1**) eluted peaks of standards, Mms6 and Ubi-Mms6; (**2**) standard curve derived from (**1**) with a vertical green line showing the elution position of Ubi-Mms6, *M*_W_: molecular weight, Ve/Vo = elution volume/void volume; (**3**) gel electrophoretic profile of Ubi-Mms6 prior to loading and in the peak eluted at ~15 mL std: molecular weight standard proteins; input: Ubi-Mms6 sample loaded on the column, peak, sample from the Ubi-Mms6 peak as identified in the A280 profile (A1); (**B**) time courses of reductase activity in the presence of 350 µM ferric citrate in 20 mM Tris, 100 mM KCl, pH 7.5 with 100 uM each of NADH and FAD with 0.8 mM ferrozine. For each binding isotherm, the data from three independently performed experiments, each normalized to a midpoint value, were averaged, and similarly treated data in the presence of each protein but no ferric citrate was subtracted. The error bars are the square root of the sum of the squares of the errors with and without ferric citrate; (**C**) the amount of iron bound per µmole of protein in 20 µM ^55^Ferric citrate was determined as described in Materials and Methods and expressed as a proportion of the amount bound by Mms6. The asterisks denote a p value of <0.05 (*) in a paired comparison with Mms6.

## Data Availability

The data that support the findings of this study are available from the corresponding author upon reasonable request.

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
