# Peer review of "The Magnetosome Protein, Mms6 from Magnetospirillum magneticum Strain AMB-1, Is a Lipid-Activated Ferric Reductase"

_ijms, 2022, doi:10.3390/ijms231810305_

Round 1

Reviewer 1 Report

In their paper “The Magnetosome Protein, Mms6, is a Lipid-Activated Ferric Reductase ” Dilini Singappuli-Arachchige and co-authors hypothesize and proof the idea about protein MmS6 role in the process of magnetosome formation. It is known, that magnetite crystals contain Fe in two oxidation states (Fe2+ and Fe3+) but MmS6 binds only Fe3+. This means that this protein itself is able to catalyze the reduction of Fe3+ to Fe2+ or “works” in tandem with another reductase. During the study, the authors clearly demonstrated the bi-functional activity of Mms6, namely the ability to bind ferric ions of the C-terminal domain and the iron-reductase activity of the N-terminal domain. They also showed that reductase activity depends on the protein localization and it enhances upon MmS6 incorporation into membranes.

The study is complex in nature it approached the question from different angles. The authors conducted comprehensive experimental studies. The presentation of the Discussion” section, in the form of several thesis greatly helps the reader to summarize the extensive information presented in the paper. The results obtained are fundamentally new and are beyond doubt.

Some technical remarks:

Line 95 (Fig.1 legends) – “....Mms6 – should be “m3Mms6”?

Line 143 – “....kcat as 2.0 x 10-5 ± 2.6 x 10-5 sec-1...”confuses the spread of the kcat value - is this normal?

Line 99, line 146 – “.. 70 uM”, “14 uM” should be “70 μM” “14 μM” ?

Line 244 – “....Twenty uM Mms6 ..should be “ Twenty μM Mms6”?

To summarize, the paper is undoubtedly of interest for specialists and can be accepted for publication as it is or after minor corrections.

Author Response

In the response below, we have colored the reviewer's comments in red and ours in italicized black:

Some technical remarks: Line 95 (Fig.1 legends) – “....Mms6 – should be “m3Mms6”? Line 99, line 146 – “.. 70 uM”, “14 uM” should be “70 μM” “14 μM” ? Line 244 – “....Twenty uM Mms6 ..should be “ Twenty μM Mms6”?

We thank the reviewer for a careful review of our manuscript and for identifying the errors listed. These errors are corrected in the revised manuscript, and we also performed a careful review of the manuscript and supplementary files to correct a few remaining errors.

Line 143 – “....kcat as 2.0 x 10-5 ± 2.6 x 10-5 sec-1...” – confuses the spread of the kcat value - is this normal?

We agree that the activity of different preparations spreads over a wide range of kcat values. To be true to the data, we decided to show all values of kcat in table S2 of the Supplementary data file and to calculate the average of all values after discarding the highest and lowest values, which we reported in the text of the manuscript. In reviewing this data in response to the reviewer’s question, we found that the calculator for the average and standard deviation of all kcat values had missed some values and thus the calculation was incorrect. The corrected average kcat is twice what we initially reported (4.8 x 10-5 ± 5.2 x 10-5 sec-1 instead of 2.0 x 10-5 ± 2.6 x 10-5 sec-1). Although a higher value than first estimated for the kcat, this value remains small and is still consistent with all conclusions reached in the manuscript.

In addition to responding to the specific questions of the reviewers, we have revised some of the text for more clarity and improved figure 1 by identifying the conditions (aerobic or anaerobic) for each plot and have changed out the plot in Fig. 1D with data on the anaerobic activity that is a better comparison with the data for aerobic activity (Fig. 1A).

Reviewer 2 Report

The paper describes data supporting the activity of the MMs6 protein from Magnetospirillum magneticum stain AMB-1 as NAD(P)H/FAD-dependent Fe(III) reductase in vitro.

The data have some merit, but the presentation does not have enough rigor.

In addition the analysis of the Kinetic data need better tools specially regarding the assessment of the goodness of fit. There are also several inconsistencies in the values and plots reported.

The modeling of the three-dimensional structure is bases on outdated tools, so this part must be reconsidered and requires more work with state-of-the-art AI software, which fortunately is currently available and does not require experience as a programmer.

Author Response

In the response below, we have colored the reviewer's comments in red and ours in italicized black:

In addition the analysis of the Kinetic data need better tools specially regarding the assessment of the goodness of fit.

We appreciate the reviewer’s concern for rigor in curve fitting and have refit all data for which we reported Kd and Km values using Sigmaplot. The results are often the same and are consistently within 5% of the values that we reported from Excel Solver, but because we now report some of the statistical output from Sigmaplot for each fit, we have converted the values on Fig. 2 and Figure S4 to those reported by Sigmaplot and added the standard error calculated by Sigmaplot to each reported Km. We have also added a value for error of the estimate for the curve fitting along with the r2 estimates already reported. The same r2 estimates were reported by both fitting programs for each dataset.

There are also several inconsistencies in the values and plots reported.

We thank the reviewer for noticing discrepancies. We consequently screened the manuscript for discrepancies and found a discrepancy in calculations of kcat compared with the numbers originally reported in the supplementary file and corrected both. Perhaps this was what the reviewer had noted. As stated in response to reviewer I, both the text in the manuscript and the information presented in the supplementary table have been corrected.

Another possible observed discrepancy is in the absolute values of the Vmax in sections B and C of figure 6. We have pointed out that different preparations and even different tests of the activity give a range of Vmax/kcat. Figure 6B and 6C and the results of different experiments in which different concentrations of lipids were used and different preparations of Mms6. We have updated the legend to this figure to clarify the difference in lipid concentrations for the two experiments.

The modeling of the three-dimensional structure is bases on outdated tools, so this part must be reconsidered and requires more work with state-of-the-art AI software, which fortunately is currently available and does not require experience as a programmer.

In figure 4, we showed the results of two programs for modeling the structure of Mms6. SWISS-PROT has embedded the AI-informed Alpha-fold program in its modeling algorithm. The second program, I-Tasser was also used for several reasons. In addition to creating a proposed structural mode, it provides a list of homologs which helped to point in us in the direction of the reductase family. The I-Tasser model shows the structure as it was modeled against the known structure of homologous reductases. Also, SWISS-PROT only shows the modeled sections of a protein for which there is high probability of an accurate prediction. However, we did not discuss these issues adequately in the text and have decided to just show the Alphafold2 model in Figure 4 because it shows the entire predicted structure. We have also added another Supplementary figure, S7, with examples of all the modeling studies that we performed in analyzing the probable structure to compare the results of several available modeling platforms. As the review can see, all but one modeling algorithm predicts a similar structure for Mms6 to that predicted by Alphafold2.

Due to the addition of the reference to Alphafold in the discussion, we also added two references to the algorithm as requested on the website. Unlike other changes, these switches of reference numbers after reference 29 do not show in “track changes”.

Round 2

Reviewer 1 Report

The paper has become much clearer after revision and can be published as it is, or after minor corrections.

Some technical remarks:

Line 109 - “E)” should be replaced with “D)”

 Lines 278-280 - (optional) The meaning of the abbreviations DMPC and POPC does not need to be given here, it is presented in the section Materials and Methods

Author Response

Thanks to the reviewer for catching the error on line 109, which has been corrected. We also agree with the reviewer that the legend to figure 6 can be much reduced by the removal of the chemical names, which we have done. However, we have also added a reference in the legend to the Materials and Methods section for this information. 

Reviewer 2 Report

The authors did make extensive changes and have addressed my previous concerns.

I can now recommend this contribution for publication.

One minor detail to be checked:

Lines 216-217 state "It is possible that, when iron is bound by the C-terminal domain, the structure becomes more ordered".

However, a number of other possibilities can be true: (i) AlphaFold 2.0 (AF2) simply did fail to produce a realistic prediction; (ii) the form of the aggregated depends on pH and AF2 does consider only one ionic form; (iii) the protein may have several forms of aggregation and AF2 did spot only one; (iv) the aggregates predicted by AF2 can assemble into aggregates of higher order, of currently unknown structure and beyond AF2 present prediction capabilities; and some others that have escaped to me. Unfortunately, there is feeble evidence to support any of them.

The statement may be simply removed or replaced by something like: "The reason behind the lack of agreement of the current AlphaFold 2.0 predictions with the experimental data is unclear at the moment."

Author Response

We are in complete agreement with the reviewer's assessment that there are many possible reasons that could explain the discrepancy between the predicted structure of the C-terminal domain and the experimental data and no solid logic to choose one over the other. After consideration, we decided to delete the identified sentence rather than to replace it with a sentence that does not add to our understanding of the system. Thanks to the reviewer for noting this weak statement. 

Round 3

Reviewer 2 Report

The MS has publication quality.

Author Response

We appreciate the reviewer's suggestion that this manuscript is of publication quality and their previous detailed and extensive review of our manuscript.